# A global synthesis and assessment of free-ranging domestic cat diet

Christopher A. Lepczyk ®[1] ✉, Jean E. Fantle-Lepczyk ®[1], Kylee D. Dunham[2,7], Elsa Bonnaud[3], Jocelyn Lindner[4], Tim S. Doherty ®[5] & John C. Z. Woinarski[6]

Free-ranging cats (*Felis catus*) are globally distributed invasive carnivores that markedly impact biodiversity. Here, to evaluate the potential threat of cats, we develop a comprehensive global assessment of species consumed by cats. We identify 2,084 species eaten by cats, of which 347 (16.65%) are of conservation concern. Islands contain threefold more species of conservation concern eaten by cats than continents do. Birds, reptiles, and mammals constitute ~90% of species consumed, with insects and amphibians being less frequent. Approximately 9% of known birds, 6% of known mammals, and 4% of known reptile species are identified in cat diets. 97% of species consumed are <5 kg in adult body mass, though much larger species are also eaten. The species accumulation curves are not asymptotic, indicating that our estimates are conservative. Our results demonstrate that cats are extreme generalist predators, which is critical for understanding their impact on ecological systems and developing management solutions.

Since house cats (*Felis catus*) were domesticated over 9000 years ago, humans have introduced them across much of the world[1]. Today, cats inhabit all continents, except Antarctica, and have been introduced to hundreds of islands[2], making them amongst the most widely distributed species on the planet. Because of this cosmopolitan distribution, cats have disrupted many ecosystems to which they have been introduced[3]. Specifically, cats spread novel diseases to a range of species including humans[4,5], out-compete native felids and other mesopredators[6], threaten the genetic integrity of wild felids[7], prey on native fauna[8], and have driven many species to extinction[9,10]. As a result, free-ranging cats (i.e., owned or unowned cats with access to the outdoor environment[11]) are amongst the most problematic invasive species in the world[12].

One attribute that has allowed cats to be successful invaders is their generalist diet. Cats are opportunistic predators and obligate carnivores[13,14] that can survive on preformed and metabolic water in food for months[15]. Furthermore, cats have evolved to survive only on animal tissue and have a set of specific nutritional adaptations as

carnivores. Specifically, cats have a limited ability to regulate enzymes of amino acid metabolism, and an inability to use plant material for conversion to amino acids and vitamins[16]. Hence, while cats consume plant material[17,18], they are dependent on meeting their energetic demands through consuming a high protein diet. As a result of these physiological needs and behavioral attributes, cats are known to depredate and scavenge a wide variety of animals[19,20].

Dietary analyses have been carried out for cats around the world for well over 100 years[21], with many studies revealing that either birds or small mammals are the dominant prey items, often depending upon the ecosystems in which the studies were conducted[13,22,23]. Such dietary differences across studies are likely in part a reflection of differences among locations in prey availability[24]. Hence, while widely distributed species are commonly found in cat diets [e.g., house mouse (*Mus musculus*), house sparrow (*Passer domesticus*)], this is more a reflection of study location and prey distribution and abundance, rather than diet preference[25]. However, recent findings suggest cats continue to

[1]College of Forestry, Wildlife and Environment, Auburn University, Auburn, AL 36849, USA. [2]Department of Biological Sciences, Nunavut Wildlife Cooperative Research Unit, University of Alberta, Edmonton, AB T6G 2R3, Canada. [3]Ecologie, Systématique et Evolution, Université Paris-Saclay, CNRS, AgroParisTech, Ecologie Systématique Evolution, 91190 Gif-sur-Yvette, France. [4]134 Moturata Rd, Taieri Mouth 9091, New Zealand. [5]School of Life and Environmental Sciences, The University of Sydney, Sydney, NSW 2006, Australia. [6]Research Institute of the Environment and Livelihoods, Charles Darwin University, Casuarina, Northern Territory 0909, Australia. [7]Present address: Cornell Lab of Ornithology, Cornell University, 159 Sapsucker Woods Road, Ithaca, NY 14850, USA. ✉e-mail: lepczyk@auburn.edu

hunt particular species of prey, even when these prey species become scarce, and that they can exhibit individual variation in hunting behavior[26]. Furthermore, cats scavenge carcasses of animals[27–29], including animals larger than they can kill, and consume spoiled and wasted food left by people[17,30], which allows them to exploit these resources and exist in a wide range of ecosystems and potentially at greater densities. However, scavenging is not the dominant source of food for cats due to their high energetic needs[31].

While previous work has focused on assessing cat diets within a specific continent (e.g., Australia[20]), ecosystem type (e.g., islands[19]), or taxonomic group (e.g., bats[32]), to our knowledge there has been no previous global attempt to comprehensively document the complement of species consumed by cats. As a result, we lack knowledge at global scale of the degree to which cats consume different animal taxa, including species of conservation concern, whether there are any taxa that cats avoid consuming, and if cats have diet selectivity based on prey or scavenge size. Here we fill these knowledge gaps about a globally distributed invasive species by constructing and evaluating the largest database of cat diet to date. Specifically, we quantified the diet of free-ranging domestic cats throughout the world by taxonomic group, island versus continental location, conservation status, body masses of diet species, and the approach used to document the dietary items.

## Results

### Taxonomic and geographic patterns of diet

Of 544 studies, 533 met our criteria for species-level data and included 2084 species eaten by cats. Notably, these 2084 species provide a conservative estimate of cat diet based on species accumulation curves, indicating that as more studies are conducted, we will discover many more prey species (Fig. 1). We found that the number of published studies on cat diet has increased dramatically over the past century (Fig. 1), with most studies being conducted in Australia (including nearby islands; $n = 215$) or North America ($n = 144$; Table 1). Consequently, studies conducted in these two regions also contributed the highest number of species known to be eaten by cats (Table 1). Globally, 288 studies were conducted on islands and 237 on continents and included records of 797 and 1,253 species, respectively (Figure SI1).

Of the individual species depredated or scavenged by cats, birds comprised 47.07% (981 species), followed by reptiles (463 species, 22.22%), mammals (431 species, 20.68%), insects (119 species, 5.71%), and amphibians (57 species, 2.74%; Fig. 2). We collapsed the 6 remaining classes (Actinopterygii, Arachnida, Chilopoda, Diplopoda, Gastropoda, Malacostraca; $n = 33$ species) into "other" representing 1.58% of the overall species tally (Fig. 2, Table SI1).

By continent, the general trend in taxonomic patterns of bird species being the most common prey, followed by reptiles, was consistent across Africa, Asia, and Australia (Fig. 2B). In Antarctica (see *Methods*), Europe, North America, and South America, the second most common prey item was mammals (Fig. 2B), though Antarctica only included birds and mammals. Notably, insects were the third most common prey item in Africa and the "other" category was tied for third most common prey with reptiles in South America.

The most commonly observed prey items included house mouse (*Mus musculus*, $n = 158$ studies, 29.64%), European rabbit (*Oryctolagus*

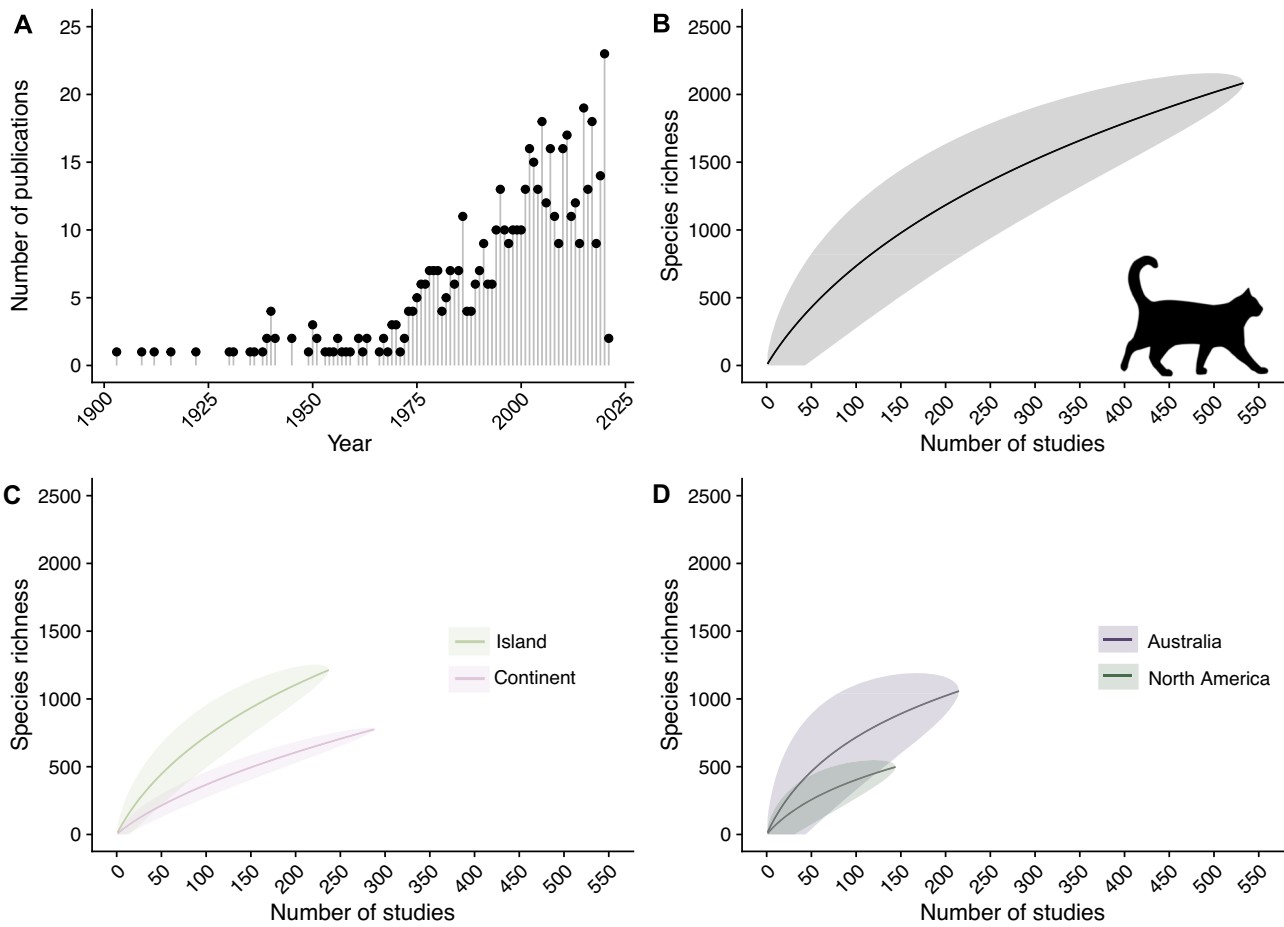

**Fig. 1 | Richness of species consumed by cats (*Felis catus*) around the world.** Number of publications describing species consumed by cats over time (**A**) and species richness of fauna identified with an increasing number of studies (**B, C**). Species accumulation curve generated for the whole data set (**B**), a subset of data for studies conducted on islands and continents (**C**), and for a subset from the two continents with the highest number of studies (**D**). Species accumulation curves include the mean (bold line) and 95% confidence intervals (shaded band). Cat icon is from http://phylopic.org and is available in the public domain.

*cuniculus*, *n* = 114 studies, 21.39%), black rat (*Rattus rattus*, *n* = 74 studies, 13.88%), house sparrow (*Passer domesticus*, *n* = 58 studies, 10.88%), and brown rat (*Rattus norvegicus*, *n* = 54 studies, 10.13%) (Table SI2).

## Impacts on species of conservation concern

A total of 347 (16.65%) cat-consumed species were of conservation concern, listed as Near Threatened, Threatened (i.e., Vulnerable, Endangered or Critically Endangered), or Extinct on the IUCN Red List (Fig. 3), with these tallies including 7.1% of the world's birds of conservation concern, 4.9% of mammals, and 2.7% of reptiles (Table 2). Globally, 25.22% of species identified on islands were species of conservation concern, whereas only 8.62% of species identified on continents were species of conservation concern (Figure SI1). Across continents/nearest continents, most species eaten are classified as Least Concern (61.99% in Africa to 86.30% in Europe; Fig. 3). Amongst the 347 species of conservation concern, birds were the greatest number of species in all categories followed closely by mammals and reptiles (Table 2). We found records of cats consuming 11 species from Australia, Mexico, the United States of America, and New Zealand that

have since been listed as extinct in the wild (EW) or extinct (EX), many of which were island endemics (e.g., Hawaiian crow [*Corvus hawaiiensis*]; Table 3). We note that 155 species, or 7.44% of the total number of species in our cat dietary compilation have not yet been evaluated under IUCN and most of these species (*n* = 108) were insects. Further, we note that the proportion of "Not Evaluated" prey species is relatively high for Africa (17.54%) and Asia (10.64%).

## Body mass of species in the diet

The median adult body mass of vertebrate species reported in cat diet was 45.45 g for all species, 13.67 g for amphibians, 62.42 g for birds, 53.22 g for mammals, and 21.35 g for reptiles (Fig. 4). Notably, cats eat prey across nearly the full range of sizes (Fig. 4). The smallest vertebrates eaten in each taxonomic class were the common dwarf skink (*Menetia greyii*; 0.47 g), Southern brown tree frog (*Litoria ewingii*; 1.69 g), Etruscan shrew (*Suncus etruscus*; 2.37 g), and the ruby-throated hummingbird (*Archilochus colubris*; 3.09). The largest vertebrate species reported for each class were the American bullfrog (*Lithobates catesbeianus*; 467.80 g), emu (*Dromaius novaehollandiae*; ≈34 kg), green sea turtle (*Chelonia mydas*; ≈133 kg), and domestic cow (*Bos taurus*; ≈760 kg). We detected no obvious thresholds in body mass of the diet, although many of the largest species recorded were likely to have been depredated as juveniles or scavenged adults.

## Approach used to determine diet

Across the methods used to determine diet the greatest number of species identified were via observed predation, while inferred predation resulted in the least number of species identified (Fig. SI2). Notably, the direct observation of predation has been increasing in many studies, likely with the application of camera traps and animal-borne video. Methodological approach varied by the taxonomic group of interest, such that there was no single method that was consistently the most common way that diet was determined (Figure SI2).

## Discussion

Our study sheds light on the predatory habits of one of the world's most successful and widely distributed invasive predators. To our knowledge this study is the most comprehensive global synthesis of

**Table 1 | Number of studies and species identified by closest continent to study site**

| Closest continent | Number of studies | Number of species |
|---|---|---|
| Africa | 48 | 171 |
| Antarctica | 6 | 20 |
| Asia | 22 | 141 |
| Australia | 215 | 1058 |
| Europe | 75 | 292 |
| North America | 144 | 498 |
| South America | 25 | 89 |

There was a total of 533 studies, however, several included samples from two or more continents. Further, we identified a total of 2,084 unique species but multiple species were identified in studies conducted across several continents (e.g., house mouse identified as a prey item on all 7 continents). Thus, we do not expect these numbers to add up to the total number of unique studies or distinct species.

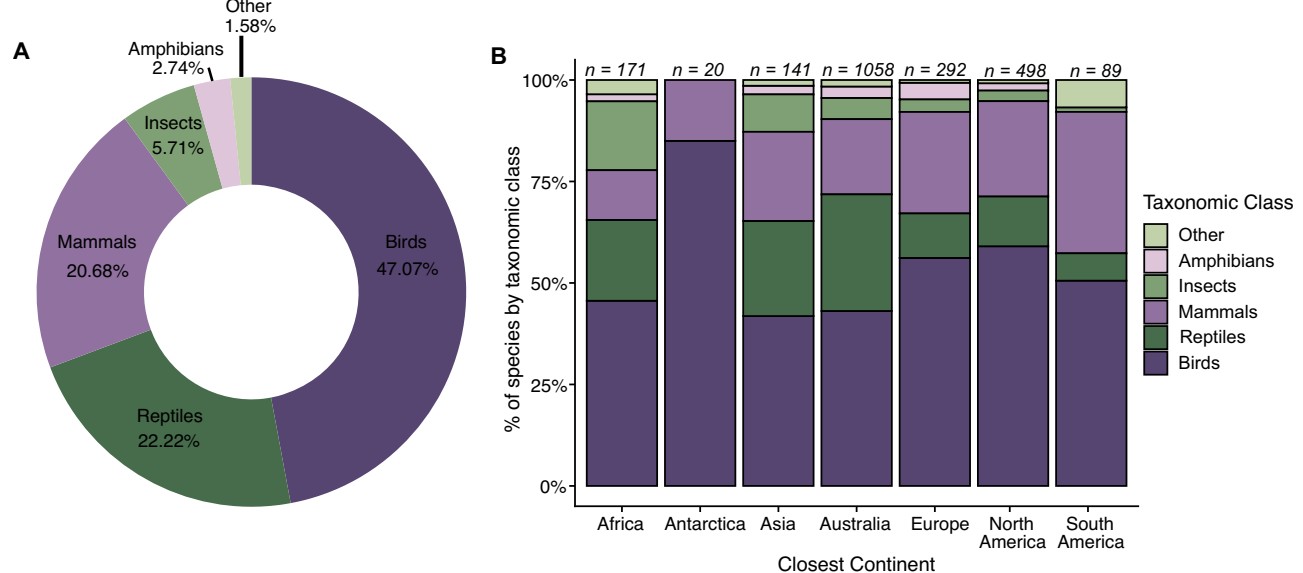

**Fig. 2 | Taxonomic patterns of species consumed by cats (*Felis catus*).** Taxonomy of the diet from an exhaustive review of cat diet studies globally (**A**) and by closest continent to the study area (**B**). The "Other" category consisted primarily of invertebrates, including, but not limited to, Arachnida, Malacostraca, Chilopoda, and Gastropoda. The numbers above each column (**B**) refer to the number of species identified on each continent. We note that these numbers do not sum to the total number of unique species identified globally because many species were identified on multiple continents.

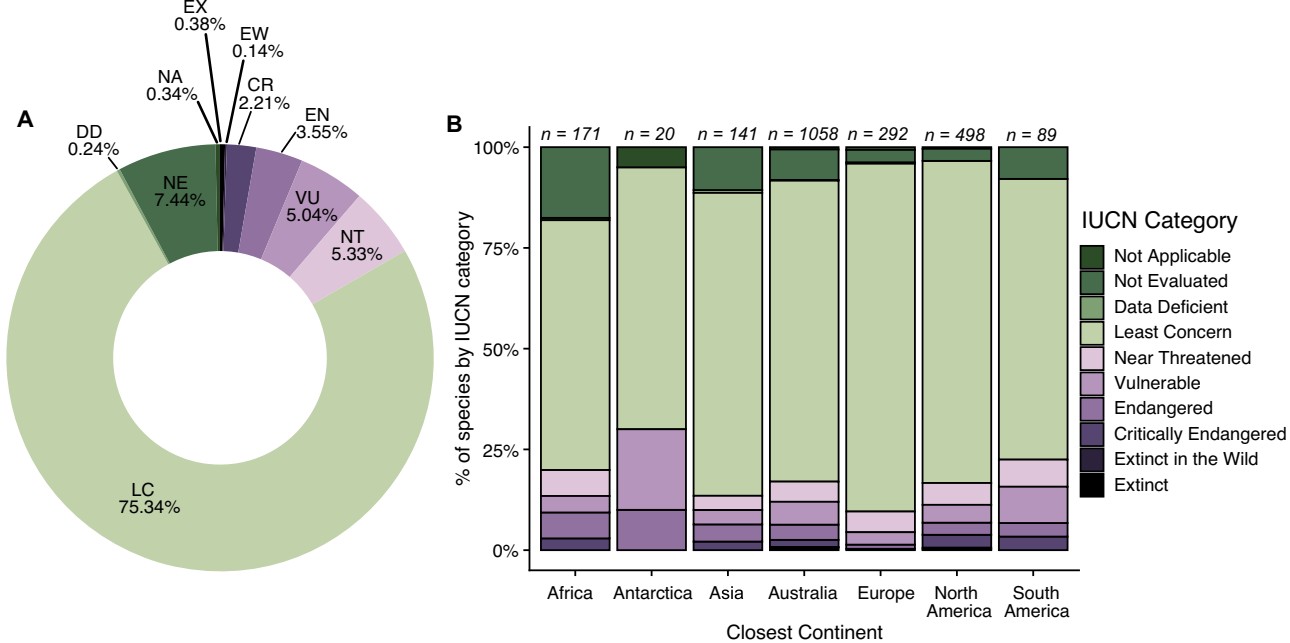

**Fig. 3 | Classification of species consumed by cats (*Felis catus*) as designated by IUCN.** Percent of species within each IUCN category identified in our review of cat diet studies globally (**A**) and by closest continent to the study area (**B**). The numbers above each column (**B**) refer to the number of species identified on each continent. We note that these numbers do not sum to the total number of unique species identified globally because many species were identified on multiple continents.

**Table 2 | Number of species identified within our sample that are of conservation concern according to IUCN by major taxonomic group**

| IUCN Classification | Birds | Mammals | Reptiles | Amphibians | Insects |
|---|---|---|---|---|---|
| NT | 68 | 24 | 17 | 1 | 1 |
| VU | 50 | 34 | 17 | 1 | 1 |
| EN | 35 | 17 | 17 | 3 | 1 |
| CR | 21 | 12 | 13 | 0 | 0 |
| EW | 2 | 0 | 1 | 0 | 0 |
| EX | 6 | 1 | 1 | 0 | 0 |
| Number of species in our sample | 182 | 88 | 66 | 5 | 3 |
| Total species of concern classified by IUCN | 2575 | 1799 | 2444 | 2965 | 3051 |
| % of IUCN species of conservation concern | 7.07% | 4.89% | 2.70% | 0.17% | 0.10% |

*NT* near threatened, *VU* vulnerable, *EN* endangered, *CR* critically endangered, *EW* extinct in the wild, *EX* extinct.
We included the percent of IUCN species of conservation concern included in the five most common taxonomic groups in our sample.

**Table 3 | Species identified as cat prey items that are classified as extinct (EX) or extinct in the wild (EW) by the IUCN**

| IUCN classification | Common name (*Scientific name*) | Locality | Country/Continent |
|---|---|---|---|
| Extinct in the Wild | Hawaiian crow (*Corvus hawaiiensis*[65]) | Island endemic, Hawaiʻi in the Hawaiian Islands | United States of America |
| | Socorro dove (*Zenaida graysoni*[66]) | Island endemic, Socorro Island in the Revillagigedo Islands | Mexico |
| | Christmas Island blue-tailed shinning-skink (*Cryptoblepharus egeriae*[67]) | Island endemic, Christmas Island | Australian territory |
| Extinct | New Zealand quail (*Coturnix novaezelandiae*[68]) | Island endemic, South Island | New Zealand |
| | Aukland merganser (*Mergus australis*[69]) | Island endemic, Auckland Islands | New Zealand |
| | Kāmaʻo (*Myadestes myadestinus*[70]) | Island endemic, Kauaʻi in the Hawaiian Islands | United States of America |
| | Chatham fernbird (*Poodytes rufescens*[71]) | Island endemic, Pitt and Mangere Islands in the Chatham Islands | New Zealand |
| | Paradise parrot (*Psephotellus pulcherrimus*[72]) | Endemic to eastern Australia | Australia |
| | Stephens Island Rockwren (*Traversia lyalli*[73]) | Island endemic, Stephens Island | New Zealand |
| | White-footed Rabbit-rat (*Conilurus albipes*[74]) | Endemic to south-eastern Australia | Australia |
| | Christmas Island Whiptail-skink (*Emoia nativitatis*[75]) | Island endemic, Christmas Island | Australian territory |

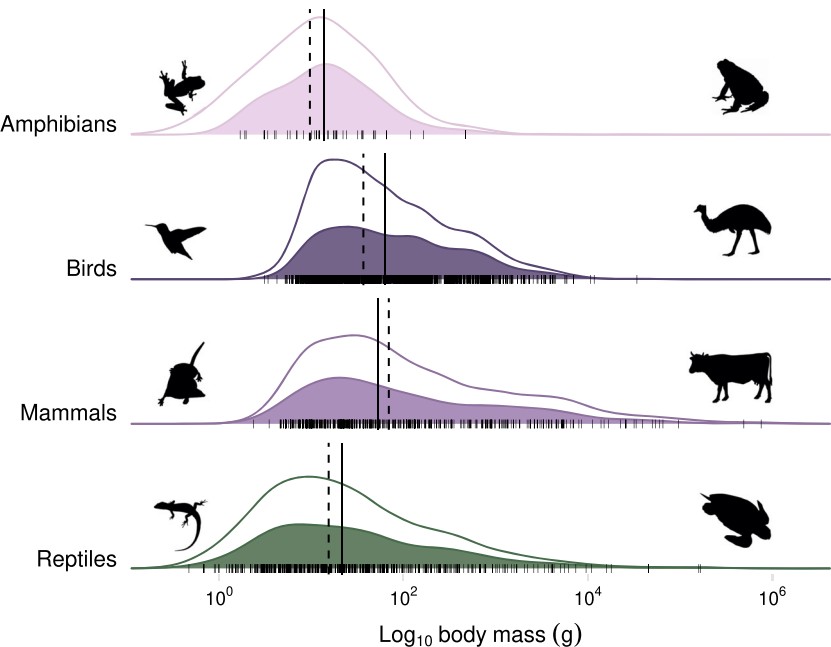

**Fig. 4 | Distributions of body masses of species consumed by cats (*Felis catus*).** Density ridge plot of $Log_{10}$ body mass (g) for amphibian, bird, mammal, and reptile species identified as being consumed by cats. The white filled distributions refer to the total distribution of known masses for species within each taxonomic class. The color filled distributions were generated from the recorded body masses of species identified in our database as being consumed by cats. Black lines below each distribution are the individual data points of the species observed in our database.

Solid vertical black lines represent the median mass for the species observed in our database and dashed vertical black lines represent the median mass for all of the known species within each taxonomic class. Species silhouettes represent the smallest and largest species consumed by cats. Species icons are from http://phylopic.org and available in the public domain or licensed under CC BY 3.0 (Emu [*Dromaius*] by Darren Naish, vectorize by T. Michael Keesey).

cat diet to date, and possibly the largest for any species worldwide. For instance, the CarniDiet database[33] contains only 823 prey/diet species across the 103 mammalian carnivores evaluated, with leopards (*Panthera pardus*) consuming the greatest diversity of species (*n* = 214). Thus, our analysis markedly increases previous estimates of the number of species depredated and scavenged by cats as well as carnivores and thus establishes a global baseline. Previous estimates of cat diet included 248 vertebrate and invertebrate species on islands worldwide[18] and 833 vertebrate species in Australia[34–38]. In total, we identified 2084 species depredated, scavenged, or otherwise consumed by cats, including nearly 9% of known birds (based on 11,162 described birds[39]), greater than 6% of known mammals (based on 6596 described mammals[40]), and approximately 4% of known reptiles (based on 11,733 described reptiles[41]). Amongst the species consumed, 347 (16.65%) were listed as Near Threatened or higher concern on the IUCN Red List. While cat diet studies were found to be globally distributed, most are from Australia and North America, with Africa, parts of Eurasia, and South America all being underrepresented. As such, there is a strong likelihood that as studies are conducted in underrepresented locations and unevaluated islands, more species of conservation concern will be added to our list.

Collectively, the finding that cats consume a large number of bird species (981 species) is unsurprising given that there are more species of birds than mammals globally (~11,000 vs. ~6500 mammals), and that birds exist in many cat-inhabited island ecosystems devoid of native mammals. Notably, there are more than 11,000 reptile species globally, but the number of reptile species recorded as eaten by cats was similar to that of mammals (463 and 431, respectively). This discrepancy may occur if cats prefer mammals and birds over reptiles, but is perhaps more likely due to sampling biases, including lower detectability of reptiles in diet samples, the small range sizes of many reptiles relative to birds and mammals[42], and that studies of reptilian prey have reported higher levels of depredation by other reptiles than

endotherms[43]. What is surprising are the relatively high number of amphibians and insects that were identified in the diet, particularly given that many studies do not have complete taxonomic resolution of insects. In particular, because of the nature of amphibian and invertebrate skin and exoskeleton, respectively, there is often far less physical material to evaluate remains in scat or digesta[35,44]. Furthermore, there is little involvement of invertebrate taxonomists in cat diet studies. What is clear from the diversity of species found in cat diets is that cats depredate and scavenge a large fraction of the species present across the range available in the landscapes they forage in and that they are representative of the distribution of all species as indicated by body mass (Fig. 4). Cats are largely eating what is present and if a species is missing in the diet analysis it is likely that the prey is either absent or rare in the surrounding environment, difficult for cats to catch and hence of low profitability, or the prey is difficult to detect (e.g., invertebrates) in scat or digesta studies. Notably, the issue of detectability may decrease in the future as molecular approaches are used in scat analysis[45].

As found in many other systematic ecological studies[46–48], there are geographic biases to our findings. Barring island systems and Australia, there is relatively little overlap with where cat diet studies have been conducted and the world's biodiversity hotspots, particularly in Africa, Asia, and South America. The lack of studies from these locations may be due to socioeconomic constraints on research effort, but possibly also lower concern about the impacts of cats on native wildlife. Islands and Australia are well studied partly due to concern about the impacts of cats on native species that are naïve to the threat of a feline predator[49,50]. In contrast, Africa, Asia, and South America have many native feline species, which may mean that i) native prey are less vulnerable to domestic cat predation due to their co-evolution with felids[51], and ii) domestic cats are less able to thrive due to competition with native carnivores[52]. Indeed, the percentage of threatened prey species is much higher on islands (25.22%) than on continents

(8.62%), perhaps due to the lack of co-evolution history between prey and predators, the high rate of endemism found globally on islands, and the disproportionately high number of threatened species restricted to islands[53]. Nonetheless, the lack of sampling in many of the most biodiverse locations on the planet suggests that we are likely missing critical locations for the evaluation of cat predation and scavenging.

While several of the most commonly found species in cat diet are cosmopolitan non-native species (e.g., *Mus musculus*, *Rattus rattus*), there was not a large set of species that were reported across most studies. In fact, most new studies on cat diet yielded hitherto unrecorded species. As a result, while small invasive rodents and birds are found in many studies, we found no indication of cats consuming just a small set of common birds and mammals, and many previous studies have reported that even in locations where cats consume a high proportion of invasive pest species, many native species are also consumed[54]. Rather, and of particular importance on islands, biodiversity hotspots, and other areas with high endemism, we found that cats preyed more generally upon whatever species were available.

A focus of previous research has been on cat predation of species of conservation concern[49,55,56]. By collating the global reports of predation and scavenging by cats, our findings create a more complete picture of the large number of IUCN Red List species, especially birds and mammals, known to be consumed, and potentially impacted, by cats. The high representation of species of conservation concern in cat diets (16.65%) is worrying given that cats have already been linked to 26% of bird, mammal, and reptile extinctions globally[9], and are recognized as major threats to many extant threatened species[9]. Since we did not look at national or regional classifications of species of conservation concern, nor at subspecies or distinct population segments, it is likely that our list contains many additional taxa of concern at local scales. Furthermore, considering that many species, especially invertebrates were Not Evaluated by the IUCN and/or were not identified in dietary studies to a level to check evaluation, additional threatened species likely exist within the diet of cats in our studies.

While the number of species our analysis documents as being eaten by cats is larger than ever previously described, all permutations of our species accumulation curves demonstrate no asymptote, indicating that additional studies are likely to continue to increase the tallies we report. This conservative estimate is likely true for a handful of reasons. First, for the purposes of this study, we only considered studies in which predation or scavenging were documented and did not record any anecdotal or secondhand reports of consumption. Second, even in studies that passed our evidentiary bar, the taxonomic resolution for many small mammals, insects, and reptiles was not consistently determined beyond genus or order due to the difficulty in identifying such species. Third, nearly all studies contained many species that were simply lumped as 'other' or 'unknown.' Fourth, large regions of the globe have not been sampled for cat predation, including many tropical regions, biodiversity hotspots, and other biodiverse locations. Fifth, different methods of collecting diet data provide different representation and biases of what cats are eating, with prey-brought-home approaches underrepresenting small prey and prey-eaten approaches less likely to record unpalatable prey[57] Sixth, as opportunistic predators, cats kill animals that they do not then consume and such fatalities are missed in nearly all reporting of cat predation. Thus, we anticipate that our database is likely to grow markedly in the future and represents only a fraction of the true magnitude of species consumed by cats globally.

Although the focus of our research was on cat diet, it builds upon over 150 years[58] of literature documenting the negative impacts that free-ranging cats pose to the environment. Aside from predation, these impacts include numerous cat-borne diseases that impact wildlife and human health and wellbeing, including toxoplasmosis, plague, and rabies[4], and in some regions (such as Australia), some of these diseases would not occur without cats. Furthermore, free-ranging cats living in clowders (aka colonies) can exacerbate these problems as well as present additional problems including excess nutrient loading, sanitation, and wildlife conflicts. Finally, simply the presence of cats outdoors can create landscapes of fear that result in changes to wildlife behavior from where a species occurs on the landscape to their foraging decisions and breeding success, which is of particular concern for threatened species[59]. Taken together, these impacts provide strong impetus to advance policy and management initiatives that seek to reduce the impacts of free-ranging cats.

Collectively, our findings demonstrate that cats are indiscriminate predators and eat essentially any type of animal that they can capture at some life stage or can scavenge. This dietary breadth lends further evidence to the myriad ways that cats can (or may) interact with native species and disrupt ecosystems because they are not dependent on any one trophic level or taxonomic group. As a result, cats are influencing a broader set of species interactions than previously understood. Ultimately, while our results are conservative, they highlight the degree to which a widely distributed invasive species is interacting with species around the world, which is critical information for furthering conservation, management, and policy work.

## Methods

We compiled information on cat diet through searches of both peer-reviewed and gray literature in Google Scholar and Web of Science using the keywords 'cat predation,' 'feral cat,' 'cat diet,' and 'Felis catus.' From each source we evaluated if it contained data on cat diet or predation, as well as reviewed its reference section for additional unique articles or databases pertaining to cat diet and predation. We iteratively evaluated any new sources identified in the references for both data and additional articles or databases in an exhaustive manner (i.e., each manuscript's references were found and evaluated for diet if they were related to cats). After this exhaustive search we identified 544 unique publications (books, journal articles, theses, and agency reports) that contained data on cat diet. Records that were not identified to the species level (i.e., only to genus) were removed from the analysis, leaving us with 533 unique publications with records of species consumed by cats. For the purposes of this research, when records were identified to the subspecies level, we reclassified them to the species level. This database contains all literature that was available through May 2021.

For each species denoted in a publication we recorded the common and/or scientific name of the species, location of the study, the method used for identifying/collecting diet data (e.g., scat analysis, gut content analysis, observed predation), year(s) of when diet/predation samples were collected, and publication citation. We updated any outdated species names and added its complete taxonomy (i.e., phylum, class), denoted whether the study location was on an island or continent, the nearest continent, and classified it according to conservation status in the IUCN Red List[60]. Note that while no species were recorded on the continent of Antarctica itself, there are islands that are geographically closer to Antarctica than other continents and as such are listed as Antarctic here. In addition, we added data on mean adult body mass for birds[61], mammals, reptiles, and amphibians[62].

To evaluate the contribution of each study to prey species richness we used species accumulation curves. Specifically, we considered each publication (i.e., study) as a unique study 'site' and the total number of unique species added per study using the vegan package (version 2.5-7[63]) in Program R (version 4.1.0[64]). However, these records were only considered as unique observations for analysis purposes if that species had not previously been identified within the database. We used the specaccum command in the vegan package (method = 'exact') to generate species accumulation curves across (1) all studies, (2) studies conducted on continents and islands (not including those

categorized as 'continent and island'), and (3) studies conducted in Australia and North America. We included species accumulation curves for Australia and North America because they comprised the majority of studies in our database. Further, we expect if the asymptote had not been met in these regions with substantially more studies than other regions, then the asymptote would not have been met on any other continent.

## Reporting summary

Further information on research design is available in the Nature Portfolio Reporting Summary linked to this article.

## Data availability

The cat diet records data used in this study will be made available upon request to the corresponding author via email. We included body mass data for birds from https://doi.org/10.6084/m9.figshare.16586228.v7 and for reptiles, amphibians, and mammals from https://doi.org/10.6084/m9.figshare.10075421.v1. Source data are provided with this paper.

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

## Author contributions

J.F.L., C.A.L., E.B., J.L., T.S.D. and J.C.Z.W. compiled the data. K.D. conducted data analysis. C.A.L., J.F.L., K.D., E.B., T.S.D. and J.C.Z.W. wrote the manuscript, and all authors reviewed the manuscript.

## Competing interests

The authors declare no competing interests.
