## [Peer Review File · Nature Communications]

A Global Synthesis and Assessment of Free-ranging Domestic Cat DietREVIEWER COMMENTS

Reviewer #1 (Remarks to the Author):

I Reviewed "A Global Synthesis of the diet of Free-ranging Cats" by Lepczyk et al. under consideration for publication at Nature Communications. This submission represents the most comprehensive accumulation of data about the diet of feral and free-ranging cats globally and represents an important synthesis for management of this invasive predator. The paper is well written, structured appropriately and the methodology and analyses are robust. The submission is timely and will be a very welcome contribution to the invasive species literature with some modifications to improve inference and present additional context. I provide my main comments below, with additional minor suggestions in the attached document. Thank you for the opportunity to review and good luck!

Cheers,

Mike Cove

The volume of species in this synthesis is impressive--Over 2,000 species! I was amazed by this number, but was curious how it compared to other small felids and to other cosmopolitan invasive predators. I reviewed the CarniDiet database (Middelton et al., 2021) to compare and was amazed that no other carnivores come close to these high of species consumption numbers. Granted, I believe that there were more in-depth diet reporting studies of feral cats and they have a broader distribution than carnivores in their native ranges and there are likely some shortcomings of the synthesis in CarniDiet being a bit less in-depth, but the highest species consumption reported by any other carnivore was 214 species consumed by common leopards--one of the most studied and broadly distributed carnivores in the world. The wildcat, from which the domestic cat is derived, was only reported to have consumed 69 species. Javan mongoose, stoat and red fox, three of the other most detrimental invasive mammalian predators in the world only consumed 5, 24, and 138 species of prey, respectively. The entire database only had 823 species of prey for all carnivores globally. These orders of magnitude difference between the other carnivores and domestic cats is very noteworthy and I provide the summaries of the CarniDiet attached as a spreadsheet for the authors to review and consider referencing to add the context of better understanding the impact of feral and free-ranging cats compared to other carnivores globally.

This dataset, when published will be very useful for researchers around the globe. I think the authors should specify a publication and accessibility plan and provide some level of updating of the database in the future, e.g., CatDiet 1.0 or something like that to make it accessible and beneficial for others in the future. I perused the data and am impressed with the volume of publications and species accumulation. The species accumulation curve did not reach an asymptote yet, which suggests the generalist nature of cats has allowed them to consume any species that they can subjugate or scavenge upon encountering. I will note that the authors acknowledge scavenging in the introduction, but do not provide enough context to understand if that is important. Does scavenging make it harder to infer cats' impact? Does this make them able to occur at high densities and strengthen impact? Does this mean they are competing with other obligate scavengers? Some qualifier of how this makes their impact on biodiversity worse or why this makes it harder to study their impacts would help readers understand their role as scavengers. I did review the data set extensively and found that there were a few observations that were easy for me to check (as author of two studies) that included species that were not in the database. These include corn snake (*Pantherophis guttatus*), Cuban tree frog (*Osteopilus septentrionalis*), and grasshopper (*Romalea guttata*) (Cove et al., 2018), and *Gambelia wislizenii* and *Sibon*

sartorii from the databases in (Schalk and Cove, 2018). I only point these out in support of the assertion that the asymptote was not reached on the species accumulation curve, particularly in cases like the two above mentioned papers, where the gut contents was an anecdote published in the discussion of Cove et al., (2018) and the others were in a Squamate predation database that we synthesized from the Herpetological Review (Schalk and Cove, 2018), which would have also been easily missed given the search criteria.

Again, I think the authors should include some commentary in the discussion about management of cats for policymakers, because there is growing evidence that this dataset supports the magnitude of impacts of feral cats, as well as pet cats and colonies of cats (e.g., Kays et al., 2020), which are becoming more and more common in legislation for cat management despite being problematic management strategies considering the volume of species consumed by cats presented herein.

One additional caveat would be to add how fear effects and killing without consumption could be missed in the current dataset, but can have very strong impacts of feral and free-ranging cats on species of conservation concern.

Title: I would recommend specifying "domestic cats" for clarity. Would it be useful to add "feral and free-ranging" to the title? I know that feral cats are inherently free-ranging, but specifying that they are part of the study population helps with context considering the magnitude of impacts of this subset of the greater free-ranging cat population. Is free-ranging the preferred term over free-roaming? I've used both and seen both in the literature and was not sure if there is a specific reason to perpetuate free-ranging over free-roaming.

Abstract: I recommend adding some context about the timing of the this submission, because there are increasing debates about the proper management of cats or lack thereof at a global scale and this synthesis certainly provides important data to support management. Possibly end with "Our findings demonstrate that cats are extreme generalist predators, which is critical for understanding their impact to ecological systems, especially when there is rising debate over their management globally."

Other comments and suggestions are included in the attached doc. Good luck!

Cove, M.V., Gardner, B., Simons, T.R., Kays, R. and O'Connell, A.F., 2018. Free-ranging domestic cats (*Felis catus*) on public lands: estimating density, activity, and diet in the Florida Keys. *Biological Invasions*, 20(2), pp.333-344.

Kays, R., Dunn, R.R., Parsons, A.W., McDonald, B., Perkins, T., Powers, S.A., Shell, L., McDonald, J.L., Cole, H., Kikillus, H. and Woods, L., 2020. The small home ranges and large local ecological impacts of pet cats. *Animal Conservation*, 23(5), pp.516-523.

Middleton, O., Svensson, H., Scharlemann, J.P., Faurby, S. and Sandom, C., 2021. CarniDIET 1.0: A database of terrestrial carnivorous mammal diets. *Global Ecology and Biogeography*, 30(6), pp.1175-1182.

Schalk, C.M. and Cove, M.V., 2018. Squamates as prey: Predator diversity patterns and predator-prey size relationships. *Food Webs*, 17, p.e00103.

Reviewer #2 (Remarks to the Author):

General comments

The manuscript provides a very interesting topic: a global analysis of animal foods consumed by free-range house cats.

The manuscript is a comparative analysis of the animal species consumed by free-ranging house cats according to 1) taxonomy and 2) IUCN categorization by continent and island versus continent, and it also analyses the log₁₀ body mass distributions of the prey species by taxon of the main food type.

The Database (Supporting Information) that forms the basis of the manuscript can be used, for example, in practical species conservation or as a basis for newer analysis aspects and hypotheses. This adds significant value to the manuscript.

Detailing the role of free-ranging house cats is somewhat one-sided. Considering that the house cat is the most popular pet, the Reader must get more comprehensive knowledge about the ecological role of free-ranging house cats. Of course, taking into account the text limitations, very concise additions are possible.

1. The global analysis of prey species according to taxonomy and threat naturally goes beyond the previous analyses carried out in smaller territorial areas (e.g. continents). However, the elaboration of the ecology function has been pushed into the background. I agree with the emphasis on the significant negative impact of free-ranging domestic cats on biodiversity, but I recommend a more complex presentation of the role.

For example, I recommend an analysis that shows the distribution of prey species by continent according to their origin (non-native versus native). The result of such an analysis would make human responsibility even more underlined. One paragraph (Lines 191-198) deals with this question in general terms.

Furthermore, it does not matter whether the species consumed is an invasive, agricultural or another pest (e.g. house mouse, *Rattus* spp., field voles, burrowing rabbits in some areas). Categorization is complex or problematic in this case. Therefore, even if it is not possible (due to time and text limitations) to reliably perform data analysis according to such a classification, it would be worthwhile to pay more attention to the issue and/or refer to the possibility of later studies in the Discussion. This would not detract from the main point.

2. It is worth giving concise but slightly more detailed information about the additional negative effects of feral or stray cats on wildlife.

Lines 45-46. In the manuscript, the authors refer, for example, to the role played in the spread of diseases. Could make a short addition about how novel diseases are spread (e.g. to native felids?). Furthermore, introgressive hybridization as a factor threatening wildcat populations throughout Eurasia and Africa should also be mentioned. In addition, hybridization (from encounters between house cats and wildcats) is also associated with the spread of diseases toward native felids.

3. Fig. 4. This is a particularly illustrative figure. However, the differences in the median mass values within and between taxa deserve further (short) analysis. Of course, the local food source and accessibility are decisive, but it is noticeable that cats eat rather the larger amphibians, birds and reptiles while consuming the smaller mammals. Taking the median values into account, is there any explanation for these differences, for example, in the behavior and profitability of prey of different sizes?

Minor comments

Title. The study is more of a comprehensive global assessment of animal species consumed by cats than a synthesis of diets.

Abstract

At the first mention, it is worth clarifying the cat taxon: free-ranging house cat (*Felis catus*).

Keywords. invasive felid, instead of invasive predator (the term predator would be a repetition)

Results

Lines 128-129. The Methods section should indicate whether all body mass data involved, including extremely outlier values (e.g. body mass of emu, green sea turtle and cattle, most likely from carrion consumption) in the analysis. Based on Figure 4, probably yes.

L. 135. Additional information can be inserted here: ... game cameras and studies based on prey-brought-home.

Discussion

L. 155-175. The different nutrient and/or energy content of each type of food could also be mentioned here.

L. 162-167. In some diet analysis methodologies, the consumption of very small amounts of arthropods is considered the prey of cat prey, so it is not listed.

L. 214. this study, instead of this manuscript

L. 232. species can (or may) interact....

Figure 1. Could present the difference between the island vs continent and Australia vs North American species accumulation curves with slightly more distinct colours.

Line 469. Instead ofspecies richness of fauna identified....: species richness of fauna based on cat foods.... (or similarly)

Figures 2-3. The caption at the top of the figure is unnecessary (% Species by IUCN Category).

Reviewer #3 (Remarks to the Author):

In this article, the authors have collated records of species killed/eaten/captured by domestic cats. They set this collation in the context of the ecological impacts of cats, primarily as these relate to predation, but also as a means of understanding how cat populations are sustained, by virtue of their generalist feeding. A broad introduction describes the diets of cats, associated predatory behaviours, and ecological impacts.

There have been several other reviews of cat diets, which are acknowledged here. This study, by contrast, is global in scope and encompasses these earlier reviews and their reviewed primary works, and adds more recent studies. But its novelty is primarily in its comprehensive approach. Its conclusions would largely have been shared with the earlier review articles and other accounts of cat predation and cat impacts.

The article does not really achieve much more than this description and collation, as useful as this latter might be. It produces some impressive statistics, that cats have been documented to eat >2000 species, and highlights this as "larger than ever previously described" (L210). It highlights variation in prey size, using this extensive database. But does not add a lot of insight beyond this tally of species, or the extraordinary generalism, opportunism, and capacity for survival shown by cats in diverse environments. These are not new observations.

The tally of species might be an impressive headline figure, but in considering it further, it is best seen as a direct consequence of the species' global range. Hence is not perhaps that

surprising. Indeed it would be more surprising if it were restricted. It is actually quite hard to tell how many or few species this is, relative to reasonable expectations. This tally represents a small proportion (4-9%) of global diversity of reptiles, birds and mammals, as reported, but what about as proportions of species within the size ranges? The ubiquity of rabbits, rats and mice, often as non-native prey "fuelling" non-native cats might be worthy of greater remark. The inclusion of domestic cows, green turtles and emus - all accorded adult masses - suggests that the survey includes a lot of scavenging.

There are some minor biases in how such data are likely to be recorded, mainly in respect of identifiability to species, which is salient in this context. But these are not a major concern, given the overriding conservation interest in vertebrates, which would be identified to species with reasonable reliability. More important biases are likely to arise in the proportion of species that are categorised as Threatened under Red List criteria, since many of the studies would have been focused on predation of such species. But again, this does not substantially detract from the main point, which is that cats eat Threatened species, and quite a lot of them. However, it is worth stipulating, as the authors will no doubt agree, records of predation alone, do not themselves provide evidence of impact, even where such a claim is credible.

Overall, this is a global collation of records that provides a useful narration to a useful data resource and provides a solid foundation for some statements that will doubtless attract public interest.

Response to Reviewer Comments

Reviewer #1:

I Reviewed “A Global Synthesis of the diet of Free-ranging Cats” by Lepczyk et al. under consideration for publication at *Nature Communications*. This submission represents the most comprehensive accumulation of data about the diet of feral and free-ranging cats globally and represents an important synthesis for management of this invasive predator. The paper is well written, structured appropriately and the methodology and analyses are robust. The submission is timely and will be a very welcome contribution to the invasive species literature with some modifications to improve inference and present additional context. I provide my main comments below, with additional minor suggestions in the attached document. Thank you for the opportunity to review and good luck!

Cheers,
Mike Cove

Response: Thank you for the positive feedback on the manuscript as well as the detailed comments which have helped to improve our manuscript.

The volume of species in this synthesis is impressive--Over 2,000 species! I was amazed by this number, but was curious how it compared to other small felids and to other cosmopolitan invasive predators. I reviewed the CarniDiet database (Middelton et al., 2021) to compare and was amazed that no other carnivores come close to these high of species consumption numbers. Granted, I believe that there were more in-depth diet reporting studies of feral cats and they have a broader distribution than carnivores in their native ranges and there are likely some shortcomings of the synthesis in CarniDiet being a bit less in-depth, but the highest species consumption reported by any other carnivore was 214 species consumed by common leopards--one of the most studied and broadly distributed carnivores in the world. The wildcat, from which the domestic cat is derived, was only reported to have consumed 69 species. Javan mongoose, stoat and red fox, three of the other most detrimental invasive mammalian predators in the world only consumed 5, 24, and 138 species of prey, respectively. The entire database only had 823 species of prey for all carnivores globally. These orders of magnitude difference between the other carnivores and domestic cats is very noteworthy and I provide the summaries of the CarniDiet attached as a spreadsheet for the authors to review and consider referencing to add the context of better understanding the impact of feral and free-ranging cats compared to other carnivores globally.

Response: Thank you for drawing our attention to CarniDiet database and providing the spreadsheet of species information. We have read over the information and agree that it is an important item to note. We have added in a new sentence at the beginning of the Discussion section noting the findings of the CarniDiet data and how our work compares to it.

This dataset, when published will be very useful for researchers around the globe. I think the authors should specify a publication and accessibility plan and provide some level of updating of the database in the future, e.g., CatDiet 1.0 or something like that to make it accessible and beneficial for others in the future. I perused the data and am impressed with the volume of

publications and species accumulation. The species accumulation curve did not reach an asymptote yet, which suggests the generalist nature of cats has allowed them to consume any species that they can subjugate or scavenge upon encountering. I will note that the authors acknowledge scavenging in the introduction, but do not provide enough context to understand if that is important. Does scavenging make it harder to infer cats' impact? Does this make them able to occur at high densities and strengthen impact? Does this mean they are competing with other obligate scavengers? Some qualifier of how this makes their impact on biodiversity worse or why this makes it harder to study their impacts would help readers understand their role as scavengers. I did review the data set extensively and found that there were a few observations that were easy for me to check (as author of two studies) that included species that were not in the database. These include corn snake (*Pantherophis guttatus*), Cuban tree frog (*Osteopilus septentrionalis*), and grasshopper (*Romalea guttata*) (Cove et al., 2018), and *Gambelia wislizenii* and *Sibon sartorii* from the databases in (Schalk and Cove, 2018). I only point these out in support of the assertion that the asymptote was not reached on the species accumulation curve, particularly in cases like the two above mentioned papers, where the gut contents was an anecdote published in the discussion of Cove et al., (2018) and the others were in a Squamate predation database that we synthesized from the Herpetological Review (Schalk and Cove, 2018), which would have also been easily missed given the search criteria.

Response: Thank you for this suggestion. As part of the manuscript acceptance at the journal is a requirement to provide the data. We are depositing the database as part of the publication and agree that designating it a first iteration is important.

In terms of scavenging, we have revised the text at lines 68-71 to read “Furthermore, cats scavenge carcasses of animals^{27,28,29}, including animals larger than they can kill, and consume spoiled and wasted food left by people^{17,30}, which allows them to exploit these resources and exist in a wide range of ecosystems and potentially at greater densities. However, scavenging is not the dominant source of food for cats due to their high energetic needs³¹.” In terms of how this affects cat densities, that is an unknown question. Scavenging most likely allows cats added opportunities to survive in many ecosystems, but the degree to which cats depend upon it or that there is enough carrion for cats to increase their numbers is unknown.

Thank you for noting several additional papers that have data and were not in our database. We agree that even using the approach we did there are items that were missed, and we agree that the fact that the species tally has not yet reached asymptote indicates that further cat dietary studies will continue to increase the pool of species known to be consumed by cats.

Again, I think the authors should include some commentary in the discussion about management of cats for policymakers, because there is growing evidence that this dataset supports the magnitude of impacts of feral cats, as well as pet cats and colonies of cats (e.g., Kays et al., 2020), which are becoming more and more common in legislation for cat management despite being problematic management strategies considering the volume of species consumed by cats presented herein.

Response: Thank you for this point. We agree that the data provide further information that will be of value to policymakers, managers, and practitioners. As a result, we have added a

new paragraph near the end of the Discussion that notes the problems of free-ranging cats and how our work further builds on that and thereby provides further evidence for the need of policies and management efforts.

One additional caveat would be to add how fear effects and killing without consumption could be missed in the current dataset, but can have very strong impacts of feral and free-ranging cats on species of conservation concern.

Response: Thank you for this point, it is quite important and something we missed in the original draft. To keep the text brief, but make sure to include this point, we have added a sentence as part of the larger paragraph in the Discussion describing the broader conservation impacts of cats as part of the previous comment.

Title: I would recommend specifying “domestic cats” for clarity. Would it be useful to add “feral and free-ranging” to the title? I know that feral cats are inherently free-ranging, but specifying that they are part of the study population helps with context considering the magnitude of impacts of this subset of the greater free-ranging cat population. Is free-ranging the preferred term over free-roaming? I’ve used both and seen both in the literature and was not sure if there is a specific reason to perpetuate free-ranging over free-roaming.

Response: Based on comments of both reviewer 1 and 2 we have modified the title to read “A Global Synthesis and Assessment of Free-ranging Domestic Cat Diet.”

Abstract: I recommend adding some context about the timing of the this submission, because there are increasing debates about the proper management of cats or lack thereof at a global scale and this synthesis certainly provides important data to support management. Possibly end with “Our findings demonstrate that cats are extreme generalist predators, which is critical for understanding their impact to ecological systems, especially when there is rising debate over their management globally.”

Response: Thanks for this point. We agree with it, but given the strict word limit of the Abstract we modified the last sentence a bit from what you suggested to read as follows “Results demonstrate that cats are extreme generalist predators, which is critical for understanding their impact to ecological systems and developing management solutions.”

Other comments and suggestions are included in the attached doc. Good luck!

Response: Thank you for the detailed comments on the manuscript. We have sought to address each one as requested. The only places that we could not fully address points were those related to word counts (e.g., Abstract is set to 150 words).

Cove, M.V., Gardner, B., Simons, T.R., Kays, R. and O’Connell, A.F., 2018. Free-ranging domestic cats (*Felis catus*) on public lands: estimating density, activity, and diet in the Florida Keys. *Biological Invasions*, 20(2), pp.333-344.

Kays, R., Dunn, R.R., Parsons, A.W., McDonald, B., Perkins, T., Powers, S.A., Shell, L., McDonald, J.L., Cole, H., Kikillus, H. and Woods, L., 2020. The small home ranges and large local ecological impacts of pet cats. *Animal Conservation*, 23(5), pp.516-523.

Middleton, O., Svensson, H., Scharlemann, J.P., Faurby, S. and Sandom, C., 2021. CarniDIET 1.0: A database of terrestrial carnivorous mammal diets. *Global Ecology and Biogeography*, 30(6), pp.1175-1182.

Schalk, C.M. and Cove, M.V., 2018. Squamates as prey: Predator diversity patterns and predator-prey size relationships. *Food Webs*, 17, p.e00103.

Response: Thank you for the additional citations. We have included several of these in the revised manuscript.

Reviewer #2:

General comments

The manuscript provides a very interesting topic: a global analysis of animal foods consumed by free-range house cats.

The manuscript is a comparative analysis of the animal species consumed by free-ranging house cats according to 1) taxonomy and 2) IUCN categorization by continent and island versus continent, and it also analyses the log₁₀ body mass distributions of the prey species by taxon of the main food type.

The Database (Supporting Information) that forms the basis of the manuscript can be used, for example, in practical species conservation or as a basis for newer analysis aspects and hypotheses. This adds significant value to the manuscript.

Response: Thank you for the positive comments.

Detailing the role of free-ranging house cats is somewhat one-sided. Considering that the house cat is the most popular pet, the Reader must get more comprehensive knowledge about the ecological role of free-ranging house cats. Of course, taking into account the text limitations, very concise additions are possible.

Response: We agree that cats are popular pets, but our focus in this manuscript is primarily centered on diet and the ecology of the species. Given the space limitations of the journal, coupled with the focus being on the species' ecology, we have provided limited edits on the point about pets in the manuscript.

1. The global analysis of prey species according to taxonomy and threat naturally goes beyond the previous analyses carried out in smaller territorial areas (e.g. continents). However, the elaboration of the ecology function has been pushed into the background. I agree with the emphasis on the significant negative impact of free-ranging domestic cats on biodiversity, but I recommend a more complex presentation of the role.

For example, I recommend an analysis that shows the distribution of prey species by continent according to their origin (non-native versus native). The result of such an analysis would make human responsibility even more underlined. One paragraph (Lines 191-198) deals with this question in general terms.

Furthermore, it does not matter whether the species consumed is an invasive, agricultural or another pest (e.g. house mouse, *Rattus* spp., field voles, burrowing rabbits in some areas). Categorization is complex or problematic in this case. Therefore, even if it is not possible (due to time and text limitations) to reliably perform data analysis according to such a classification, it would be worthwhile to pay more attention to the issue and/or refer to the possibility of later studies in the Discussion. This would not detract from the main point.

Response: Analyzing the distribution of the >2000 species would be a massive undertaking at the global scale. While we agree it would be interesting and useful to investigate the relative impacts of cats on native/non-native species, it is beyond the scope of the current work. We strongly believe an analysis at that scale warrants an independent manuscript and should not be included in this work. That said, we have added text to note the point that even when invasive species are depredated, many native species are as well.

2. It is worth giving concise but slightly more detailed information about the additional negative effects of feral or stray cats on wildlife.

Lines 45-46. In the manuscript, the authors refer, for example, to the role played in the spread of diseases. Could make a short addition about how novel diseases are spread (e.g. to native felids?). Furthermore, introgressive hybridization as a factor threatening wildcat populations throughout Eurasia and Africa should also be mentioned. In addition, hybridization (from encounters between house cats and wildcats) is also associated with the spread of diseases toward native felids.

Response: Thank you for this point, we agree that hybridization is an item we missed and have included it in the revised sentence. We also added text that cats can spread diseases to humans.

3. Fig. 4. This is a particularly illustrative figure. However, the differences in the median mass values within and between taxa deserve further (short) analysis. Of course, the local food source and accessibility are decisive, but it is noticeable that cats eat rather the larger amphibians, birds and reptiles while consuming the smaller mammals. Taking the median values into account, is there any explanation for these differences, for example, in the behavior and profitability of prey of different sizes?

Response: The median mass of eaten species for amphibians, reptiles, and birds is greater than the median mass for the total number of species (with mass data). The opposite is true for mammals. This difference is largely driven by the disproportionate number of large mammalian species in the data used to generate the full taxonomic class mass figure. Overall, >10% of mammalian species (with mass data) are >5 kg, which is roughly the size of a cat. The other taxonomic classes have <2% of species with body mass >5 kg. Across all classes

99% of the species eaten by cats were < 5 kg indicating supporting previous work that indicates cats typically eat prey smaller than themselves. The high body masses of species in the tail end of the mass distribution for mammals is likely driving the observed pattern.

Minor comments

Title. The study is more of a comprehensive global assessment of animal species consumed by cats than a synthesis of diets.

Response: We have revised the title to accommodate both Reviewer 1 and 2's suggestions to read as follows "A Global Synthesis and Assessment of Free-ranging Domestic Cat Diet."

Abstract

At the first mention, it is worth clarifying the cat taxon: free-ranging house cat (*Felis catus*).

Response: We appreciate the point, but we are already at the word limit and with the scientific name already included we did not add the word.

Keywords. Invasive felid, instead of invasive predator (the term predator would be a repetition)

Response: Done.

Results

Lines 128-129. The Methods section should indicate whether all body mass data involved, including extremely outlier values (e.g. body mass of emu, green sea turtle and cattle, most likely from carrion consumption) in the analysis. Based on Figure 4, probably yes.

Response: We included all of the body mass data. We have left the text stand as we did not remove any data from the figure.

L. 135. Additional information can be inserted here: ... game cameras and studies based on prey-brought-home.

Response: As reviewer 1 also noted a similar point we revised the sentence to indicate that the advent of camera traps is one of the main reasons. We did not add in the prey brought home element as that was not as clear of an increasing trend.

Discussion

L. 155-175. The different nutrient and/or energy content of each type of food could also be mentioned here.

Response: While energy content of species is an interesting point, it is beyond the scope of the manuscript as it would entail adding in caloric data by type (carbohydrate, lipid, protein) for each species and result in a separate analysis. We agree that this is an interesting point, but believe that to both do the analysis correctly and interpret the data would be a considerable undertaking that would essentially result in a separate manuscript. Furthermore, we did not

add text stating that different species would have different caloric values here as we believe that this point is intuitive and does not fit within the text.

L. 162-167. In some diet analysis methodologies, the consumption of very small amounts of arthropods is considered the prey of cat prey, so it is not listed.

Response: We were a bit unclear about the point raised here. If you mean that there are cats that are consuming insectivorous species and as a result there are insects inside the gut of an insectivore when a cat eats it and that an author may not have included it in their diet summary, we understand the point. While we follow the logic here, we would have no way of knowing whether an author of a study did or not include insect material that could have occurred as a result of this line of reasoning unless they stated it. That said, many papers noting insect material did not evaluate it taxonomically (hence our point in this section of the manuscript) or to species level and as a result was not included in our analysis. Ultimately, this is an interesting point, but one that cannot be assessed based on what is typically presented in the methods of each paper.

L. 214. this study, instead of this manuscript

Response: Done.

L. 232. species can (or may) interact....

Response: Done.

Figure 1. Could present the difference between the island vs continent and Australia vs North American species accumulation curves with slightly more distinct colours.

Response: The color palette is consistent across all figures and cannot be altered. However, we added the background color for the confidence intervals to the legend which makes the color differences more visible.

Line 469. Instead ofspecies richness of fauna identified...: species richness of fauna based on cat foods.... (or similarly)

Response: We didn't quite understand the suggested edit here and believe the current figure legend correctly captures the data it represents.

Figures 2-3. The caption at the top of the figure is unnecessary (% Species by IUCN Category).

Response: Done.

Reviewer #3:

In this article, the authors have collated records of species killed/eaten/captured by domestic cats. They set this collation in the context of the ecological impacts of cats, primarily as these relate to predation, but also as a means of understanding how cat populations are sustained, by virtue of

their generalist feeding. A broad introduction describes the diets of cats, associated predatory behaviours, and ecological impacts.

There have been several other reviews of cat diets, which are acknowledged here. This study, by contrast, is global in scope and encompasses these earlier reviews and their reviewed primary works, and adds more recent studies. But its novelty is primarily in its comprehensive approach. Its conclusions would largely have been shared with the earlier review articles and other accounts of cat predation and cat impacts.

Response: We concur with the reviewer that there have been two to three broad-scale reviews of cat diet, but would disagree that the work presented in this manuscript and its conclusions could have been deduced from prior research. Our findings present not only a global accounting of the taxa that cats eat, but evaluate it by taxonomic group, IUCN status, body mass, geographic location, and the fraction of known species that cats consume. In fact, our work demonstrates that cats are by far the most generalist predator ever described by scientists, which is important both for basic ecology as well as for conservation. Finally, as reviewer 1 notes, the results here dramatically change our knowledge about carnivores as well as the significant challenge that cats pose to conservation. Overall, we believe our study is a compelling set of findings, based on the largest assembled dataset, that will be widely read, cited, and used.

The article does not really achieve much more than this description and collation, as useful as this latter might be. It produces some impressive statistics, that cats have been documented to eat >2000 species, and highlights this as “larger than ever previously described” (L210). It highlights variation in prey size, using this extensive database. But does not add a lot of insight beyond this tally of species, or the extraordinary generalism, opportunism, and capacity for survival shown by cats in diverse environments. These are not new observations.

Response: As we noted in the previous point, we disagree with the reviewer’s assertions that the manuscript does not provide much more than a summary or highlight, noting particularly our consideration of assessment by taxonomic group, IUCN status, body mass, geographic location, and the fraction of known species that cats consume. The fact is that cats are a globally distributed non-native species that have been evaluated in many separate contexts, but not in a unified manner as presented here. If we are to understand cat natural history and ecology as well as design conservation and management strategies to address free-ranging cats, the work presented here is critical.

The tally of species might be an impressive headline figure, but in considering it further, it is best seen as a direct consequence of the species’ global range. Hence is not perhaps that surprising. Indeed it would be more surprising if it were restricted. It is actually quite hard to tell how many or few species this is, relative to reasonable expectations. This tally represents a small proportion (4-9%) of global diversity of reptiles, birds and mammals, as reported, but what about as proportions of species within the size ranges? The ubiquity of rabbits, rats and mice, often as non-native prey “fuelling” non-native cats might be worthy of greater remark. The inclusion of domestic cows, green turtles and emus - all accorded adult masses - suggests that the survey includes a lot of scavenging.

Response: Again, we would disagree with the reviewer's views here. First, there are numerous non-native species that have been globally introduced by humans that do not consume the variety of species identified in the manuscript. To date, there is no evidence that free-ranging dogs have the same degree of diet breadth, or rat species, or the numerous bird species. Barring conducting the exact same review with other non-native species, the literature does not demonstrate any other globally distributed non-native species as having such a diet breadth. Moreover, the current work not only far surpasses any previous estimate, but shows no evidence that we have identified many diet species as the species accumulation curves do not come close to leveling off. Second, there is no generalized formula upon which to reasonably expect a certain number of prey species to be found in the diet. While interesting, that is beyond the scope of this analysis and would warrant a more general evaluation of multiple species dietary patterns. Third, the proportions of mammals and birds consumed by cats is not small when considering both the number of individuals those represent and the fact that the species accumulation curves do not level off. Demonstrating that one species consumes such a large number of different species is of significant importance to both ecology and conservation. Fourth, the presence of other non-native species in the diet may simply reflect their larger cosmopolitan distributions over any preferences. Most studies are not recording abundance and thus it is beyond the scope of the database to make inferences on whether or not these non-native dietary species differ in amounts. Notably, cats need animal-based protein on a regular basis and whether the food is native or non-native is not a driving factor of predation. Finally, yes there are large bodied animals in the diet, some of which are most certainly scavenged, but others that may be depredated when they are smaller-bodied nestlings or juveniles, eggs, etc. As we note in the caveats to the research, many studies do not allow for discrimination between scavenging and predation. Thus, while scavenging certainly is likely to increase as body size increases, we do not have any data demonstrating where that threshold may exist.

There are some minor biases in how such data are likely to be recorded, mainly in respect of identifiability to species, which is salient in this context. But these are not a major concern, given the overriding conservation interest in vertebrates, which would be identified to species with reasonable reliability. More important biases are likely to arise in the proportion of species that are categorised as Threatened under Red List criteria, since many of the studies would have been focused on predation of such species. But again, this does not substantially detract from the main point, which is that cats eat Threatened species, and quite a lot of them. However, it is worth stipulating, as the authors will no doubt agree, records of predation alone, do not themselves provide evidence of impact, even where such a claim is credible.

Response: We concur with the reviewer, but note that the manuscript does not focus on population level impacts, only the diet.

Overall, this is a global collation of records that provides a useful narration to a useful data resource and provides a solid foundation for some statements that will doubtless attract public interest.

Response: Thank you for the positive assessment.

Reviewers' Comments:

Reviewer #1:

Remarks to the Author:

I have reviewed the revised manuscript " A Global Synthesis and Assessment of Free-ranging Domestic Cat Diet" by Lepczyk et al. And the authors have thoroughly and sufficiently addressed all of my comments on the previous manuscript draft. This will be an excellent contribution to the Invasive Species literature! It's a great baseline of the growing understanding of feral and free-ranging cat impacts on biodiversity. Thank you for the opportunity to review and help fine tune such a worthwhile endeavor.

Cheers,

Mike Cove

Reviewer #2:

Remarks to the Author:

The authors have made appropriate corrections to the manuscript where it was necessary. I received an appropriate response to my comments. The length of the text is strictly limited; I agree and accept that the authors did not deal with some questions or suggestions within the manuscript that were significantly outside the main scope of the study.

Excellent study!

József Lanszki